# Postprandial transfer of colostral extracellular vesicles and their protein and miRNA cargo in neonatal calves

**Benedikt Kirchner**[1]*, **Dominik Buschmann**[1,2], **Vijay Paul**[3], **Michael W. Pfaffl**[1]

**1** Division of Animal Physiology and Immunology, TUM School of Life Sciences Weihenstephan, Technical University of Munich, Munich, Germany, **2** Institute of Human Genetics, University Hospital, LMU Munich, Munich, Germany, **3** National Research Centre on Yak, ICAR, Dirang, India

* benedikt.kirchner@wzw.tum.de

**Data Availability Statement:** Raw sequencing reads were deposited in the European Nucleotide Archive (ENA) under the accession number PRJEB28002 (http://www.ebi.ac.uk/ena/data/view/PRJEB28002)

## Abstract

Extracellular vesicles (EVs) such as exosomes are key regulators of intercellular communication that can be found in almost all bio fluids. Although studies in the last decade have made great headway in discerning the role of EVs in many physiological and pathophysiological processes, the bioavailability and impact of dietary EVs and their cargo still remain to be elucidated. Due to its widespread consumption and high content of EV-associated microRNAs and proteins, a major focus in this field has been set on EVs in bovine milk and colostrum. Despite promising *in vitro* studies in recent years that show high resiliency of milk EVs to degradation and uptake of milk EV cargo in a variety of intestinal and blood cell types, *in vivo* experiments continue to be inconclusive and sometimes outright contradictive. To resolve this discrepancy, we assessed the potential postprandial transfer of colostral EVs to the circulation of newborn calves by analysing colostrum-specific protein and miRNAs, including specific isoforms (isomiRs) in cells, EV isolations and unfractionated samples from blood and colostrum. Our findings reveal distinct populations of EVs in colostrum and blood from cows that can be clearly separated by density, particle concentration and protein content (BTN1A1, MFGE8). Postprandial blood samples of calves show a time-dependent increase in EVs that share morphological and protein characteristics of colostral EVs. Analysis of miRNA expression profiles by Next-Generation Sequencing gave a different picture however. Although significant postprandial expression changes could only be detected for calf EV samples, expression profiles show very limited overlap with highly expressed miRNAs in colostral EVs or colostrum in general. Taken together our results indicate a selective uptake of membrane-associated protein cargo but not luminal miRNAs from colostral EVs into the circulation of neonatal calves.

## Introduction

MicroRNAs (miRNAs) are small non-coding RNAs of approximately 21 nt that can regulate gene expression post-transcriptionally by hybridizing to complementary sequences in the 3'-untranslated region of mRNAs or in their coding region [1]. High degree of complementarity between miRNA and mRNA leads to destabilization and subsequent degradation, while a

**Funding:** The author(s) received no specific funding for this work.

**Competing interests:** The authors have declared that no competing interests exist.

weaker pairing of primarily the seed region prevents translation to proteins at ribosomes [2]. Although miRNA biogenesis is a well-studied topic, resulting mature sequences display a heterogeneity far greater than originally assumed [3]. The complexity of these isoforms of miRNAs (isomiRs) stems mostly from divergent processing by DROSHA and DICER during cleaving of pri- and pre-miRNAs [4], but additional variants, diverging from pri- and pre-miRNA sequence templates, are also generated via exonucleases, nucleotidyl transferases and RNA editing [3]. isomiRs were shown to be incorporated into RNA-induced silencing complexes (RISC) [5] and achieve functional importance by cooperatively regulating common biological pathways [6]. Moreover, the potential and specificity of isomiR distribution patterns as biomarkers in development [7], cancer research [8], and even gender studies [9] were clearly demonstrated.

Although the general importance of miRNAs is undisputed, the functional relevance of dietary miRNAs still remains to be elucidated. Initial discoveries on uptake of plant miRNAs from rice [10] and honeysuckle [11] were followed by a number of studies that failed to reproduce the original findings [12–14] and were able to plausibly attribute them to contamination, poor study design or sequencing artefacts [15,16]. Even though the bioavailability of plant-derived miRNAs in food appears to be arguably refuted, it seems a special case can be made for dietary miRNAs associated with extracellular vesicles (EVs). EVs such as exosomes are key regulators of intercellular communication [17] and various studies in recent years have highlighted their importance in development [18] or promotion of disease progression [19].

One dietary source rich in EV-encapsulated miRNAs is milk [20]. Studies have shown that a large proportion of miRNAs in milk is localised in exosomes and exosome-like EVs [21]. Despite a significant reduction during processing or storage [22,23], a majority of milk miRNAs are stable and readily detected in commercial dairy products and even milk powder [24]. More importantly, miRNAs in milk EVs were proven to resist degradation under simulated gastrointestinal tract conditions [25,26], and *in vitro* experiments verified their ability to enter human intestinal cells [27], endothelial cells [28] as well as circulating immune cells [29]. Unfortunately, results of *in vivo* studies on orally administered milk miRNAs remain inconsistent. Baier *et al.* demonstrated a meaningful uptake of miRNAs after consumption of physiological amounts of bovine milk in adult humans and mice, coupled with a decrease in murine plasma levels of these miRNAs after feeding a diet depleted of bovine milk exosomes [30]. Independent replication of this study on the same samples, however, could not reproduce the results albeit this study suffered from technical problems during sample storage and EV degradation could not be excluded [31]. Well-designed experiments in knockout mice deficient in milk-enriched miRNAs failed to detect any significant postprandial miRNA increase in newborn pups after feeding wild type milk [32] and are therefore conflicting with studies that showed accumulation of orally administered fluorophore-labelled bovine milk exosomes in peripheral tissues in adult mice [33]. While efficiency of EV-associated miRNA transfer remains disputed, a systemic effect of orally administered milk EVs was proven in multiple studies including an attenuating effect on arthritis in mouse models [34]. Furthermore milk derived EVs might prove to be important players in drug delivery. Agrawal et al showed a significant tumor growth inhibition by drug loaded EVs compared to intraperitoneal injection [35] and drug delivery by milk derived EVs seems to induce less or no adverse immune and inflammatory responses [36,37].

So far, the vast majority of experiments has focused on the potential uptake of miRNAs from commercially available bovine milk in adult non-bovine species, neglecting the unique advantages of newborn calves and colostrum as a model for transfer of dietary miRNAs. It stands to reason that an uptake of milk or colostrum-derived EVs is just as likely to be observed between mother and direct offspring as a cross-species transfer to adult individuals.

Colostrum not only contains significantly higher amounts of miRNAs compared to mature milk, it was also shown that these are largely associated with EVs [20]. Moreover, transfer of colostral protein such as immunoglobulins to the circulation has already been studied extensively in ruminants. Although the intestinal uptake can be receptor-mediated, it is mostly driven by unspecific pinocytosis and occurs exclusively during the first days after parturition [38]. Furthermore, immediate postnatal colostral EVs were shown to be highly enriched for immune relevant proteins and factors involved in intestinal cell proliferation and displayed a protein composition quite dissimilar to milk EVs from later lactation stages and even colostrum EVs from day 2 or 3 [39]. While this prevents further conclusions on mature milk EVs or a potential uptake in other species, given the unique nature of colostral EVs [40], studying the effects of their very first dietary EVs on new-borne calves harbours great potential for insights in EV transfer in general.

The potential implications of a widespread absorption of dietary miRNAs are far-reaching and intimidating. Given the unique role miRNAs play in almost all physiological and pathological processes [41], it would cause no less than a paradigm shift in our perception of nutrition in general [42]. The objective of this study was to assess the potential transfer of colostral EVs and their cargo to the circulation of newborn calves by analysing colostrum-specific protein, miRNA and isomiR markers.

## Material and methods

### Sample collection

Blood and colostrum were sampled from randomly selected, healthy, multiparous, pregnant Brown Swiss cows (n = 9) housed at research station Veitshof (TU Munich, Weihenstephan) on the day of parturition. Blood samples from calves were taken directly before the first ad libitum feeding with colostrum (0 h) as well as 1 h, 3 h, 6 h afterwards plus directly before second feeding (9–12 h). All blood samples were drawn from *vena jugularis* in 9 ml K3 EDTA-Vacuette tubes (Greiner bio-one) with single-use needles (20G x 1", Greiner bio-one). Plasma and blood cells were separated within half an hour after sampling by centrifugation at 1850 g for 20 min at 4 ˚C. Comparably, colostrum was centrifuged at 1850 g for 30 min at 4 ˚C within one hour post sampling. After removal of the fat layer, skimmed colostrum and colostrum cells were collected. All samples including whole blood and colostrum were stored at -80 ˚C until further analysis. Colostrum was collected as total quarter milk and fed within 2 h of parturition, and calves were monitored to prevent autonomous feeding before blood sampling. Animals were housed and fed according to good animal attendance practice under permanent surveillance of a veterinarian, and all efforts were made to minimize suffering (permission number 55.2-1-54-2531-5-08). All animal trials were approved by the government of Upper Bavaria according to the German Protection of Animal Acts and no animals were sacrificed for this study and all animals were housed for further research.

### Isolation of extracellular vesicles

EVs were isolated from defatted and cell-free sample fractions (~ 4 ml of plasma and 66 ml of skimmed colostrum) by differential ultracentrifugation as described previously [21,43] followed by flotation into a sucrose density gradient (SDG) [43]. In short, samples were diluted in PBS if necessitated by uneven fill levels, and pre-cleared by low-speed centrifugation (12,000 g, 1 h, k-factor: 2335.3) removing any remaining fat and cell residues. Plasma EVs were pelleted from 12,000 g supernatant (100,000 g, 2 h, k-factor: 278.3). For skimmed colostrum samples, the 12,000 g supernatant was subjected to further centrifugation steps at 35,000 g (1 h, k-factor: 797.4) and 70,000 g (3 h, k-factor: 397.9) before EVs were pelleted at 100,000 g (1 h, k-

factor: 278.3). To increase EV purity, pellets were resuspended in PBS and separated on a discontinuous top-down sucrose density gradient (30%, 40%, 50%, 60%) at 200,000g (18 h, k-factor: 110.3). Sequential fractions were diluted 1:10 in PBS and washed by ultracentrifugation (100,000 g, 1 h, k-factor: 278.3). Sucrose gradient fractions of 40% and 50%, corresponding to a density of 1.1764 g/ml and 1.2296 g/ml, respectively, were pooled to increase yield, as results showed these to be most enriched in colostrum EVs (S1 Table). Resulting EV pellets were either resuspended in PBS for further characterization, or directly lysed in QIAzol (Qiagen) for RNA extraction. All centrifugation steps were carried out at 4 °C using an Optima LE-80K ultracentrifuge and a SW40 or SW60 rotor (Beckman Coulter).

## Total RNA extraction and characterization

Total RNA from blood and colostrum EVs (SDG fraction 40–50% only), cells from 1 ml of whole blood or 100 ml of colostrum as well as 1 ml of each non-fractionated fluid was isolated using the miRNeasy Mini Kit (Qiagen) according to the manufacturer's instructions. A workflow for all RNA-related procedures is provided in Fig 1. To assess RNA quantity and exclude potential contaminations, samples were analysed on a NanoDrop spectrophotometer (Thermo Fisher Scientific), and total RNA profiles were assessed on the 2100 Bioanalyzer (Agilent Technologies) via RNA 6000 Nano and Small RNA Assay (Agilent Technologies).

## Library preparation and Next-Generation small RNA sequencing

Comprehensive miRNA expression profiles of three randomly selected cows and their offspring were generated for all extracted compartments and time points (total n = 57) by small RNA-Seq. To focus on the evaluation of a potential transfer of EVs and their cargo from colostrum to the circulation, only the predominant colostral EV fractions corresponding to 40–50% SDG were sequenced. The library preparation was carried out as described previously [44–46] utilizing the NEBNext Multiplex Small RNA Library Prep Set for Illumina (New England Bio-Labs). Libraries were prepared from 100 ng total RNA except for calf plasma EV samples, where, due to their low concentration, the entire RNA yield was used. Small RNA specificity was achieved by size selection of PCR products using high resolution 4% agarose gel electrophoresis and retrieving bands corresponding to miRNA-adaptor-constructs (130–150 base pairs). Prior to 50 cycles of single-end sequencing on a HiSeq2500 (Illumina), fragment length and library purity were further confirmed by capillary electrophoresis (2100 Bioanalyzer High Sensitivity DNA Assay, Agilent Technologies). Raw sequencing reads were deposited in the European Nucleotide Archive (ENA) under the accession number PRJEB28002 (http://www.ebi.ac.uk/ena/data/view/PRJEB28002).

## Data analysis

Sequencing data was processed using a self-compiled bioinformatic pipeline as described previously [47,48] with the added functionality of discovering and quantifying isomiRs. In short, 3'-end adaptor sequences were trimmed using Btrim [49], and length distribution and sequencing quality were monitored via FastQC [50]. To prevent false positive mappings to miRNAs and isomiRs, reads matching rRNA, tRNA, snRNA or snoRNA sequences obtained from RNAcentral [51] along with reads shorter than 16 nt were excluded from further analyses. Filtered reads were then aligned to a newly designed mapping reference consisting of all bovine miRNAs (miRBase v21) [52] and their respective miRNA isoforms. isomiR sequences were derived from canonical miRNA sequences by consecutive trimming of up to 6 nt or addition of 3 nt on 5'- and 3'-end, and included mismatch information obtained during mapping.

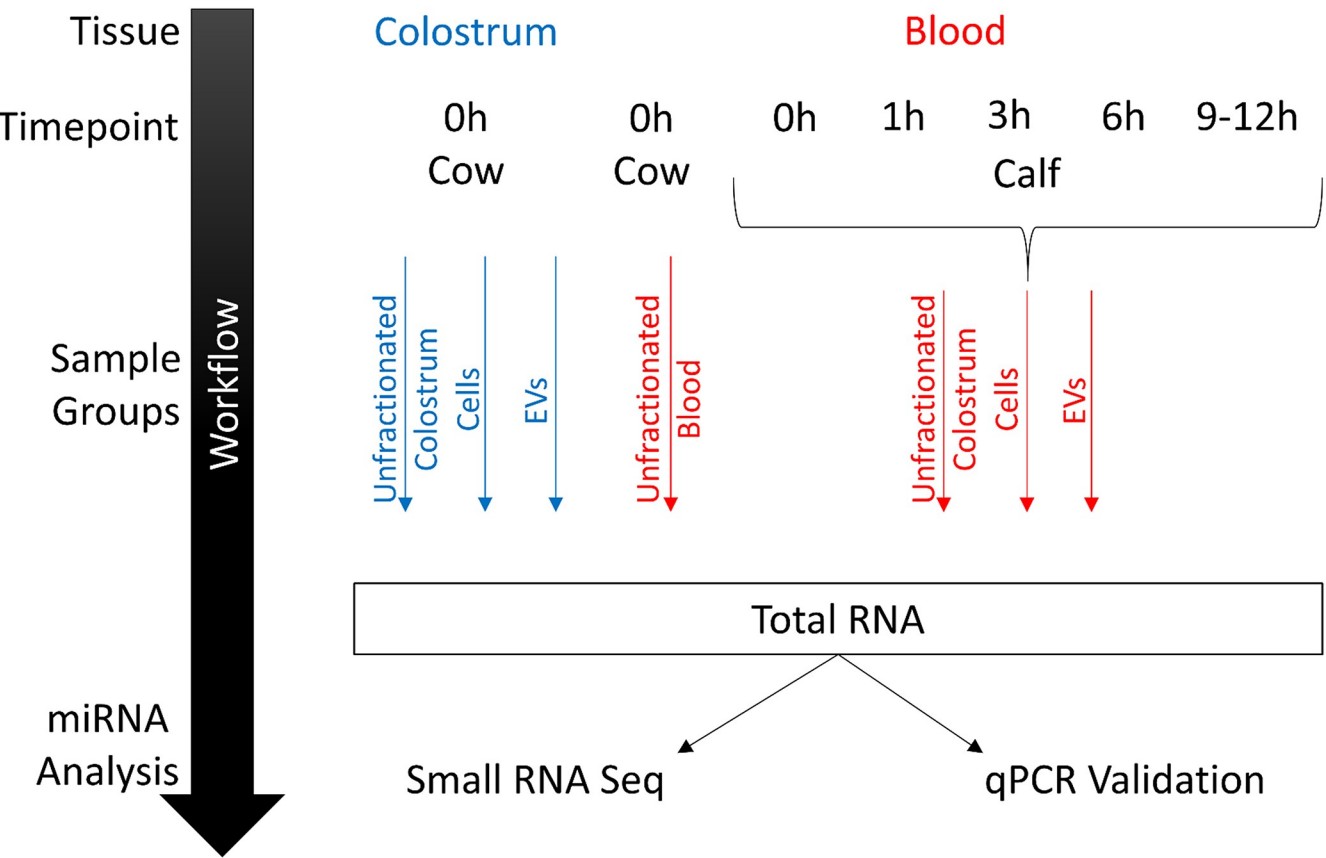

**Fig 1. Schematic overview of sample groups for RNA extraction and transcriptomic analyses.** Total RNA was extracted from unfractionated, cellular and EV compartments of colostrum and calf blood samples, as well as unfractionated blood from cows before first feeding and at four defined postprandial time points (n = 3 each) before being profiled by small RNA-Seq. Expression of differentially regulated and highly abundant miRNAs was subsequently assessed by RT-qPCR in an expanded animal cohort (cows n = 6, calves n = 8).

Alignment was performed using Bowtie [53], allowing for a single mismatch over the whole sequence, and applying the 'best' algorithm. From the resulting SAM files, isomiR read count tables were generated by incorporating mismatch information that describes potential polymorphic isomiRs together with sequence additions and trimmings, and finally calling the sum of each individual sequence. Differential gene expression profiles were obtained employing the DESeq2 package (v1.18.1) [54] from bioconductor and the included normalization, testing and false discovery correction algorithms. Expression changes of isomiRs and underlying miRNAs were considered significant if adjusted p-values were $\leq 0.05$ and $\log_2$ fold changes $\geq |1|$ combined with a minimal abundance of baseMean $\geq 50$. Similarities of isomiR expression profiles were visualised via hierarchical clustering (euclidean distances, ward's method), principal component analysis of regularised log-transformed, normalised read counts, and Venn diagrams using R (v3.4.3) [55] and relevant packages [56–59]. Potential implications of significantly up-regulated miRNAs in calf EV fractions were evaluated by enrichment analysis of KEGG pathways [60,61] following the advice from Godard and van Eyll to minimize the false positive effect of single miRNAs with multiple targets within the same pathway [62] for experimentally verified mRNA targets of human homologues with strong evidence obtained from miRTarBase [63].

### RT-qPCR

The Exiqon miRCURY LNA Universal RT microRNA PCR system (Exiqon) was used to validate miRNAs selected from NGS results in a larger cohort comprised of 6 cows and their 8 calves including two twin births (total n = 144). Reverse transcription and qPCR were performed according to the manufacturer's instructions with 10 ng of total RNA as starting input except for calf plasma EV samples, where 2 μl of undiluted sample was used due to their low yield. Assays for validation included: bta-miR-21-5p, bta-miR-26b, bta-miR-30a-5p, bta-miR-141-3p, bta-miR-144, bta-miR-146a, bta-miR-146b-5p, bta-miR-148a-3p, bta-miR-200a, bta-miR-200b, bta-miR-451 and bta-miR-2285t. Specificity of all assays and samples was ensured by using non-template and negative RT controls in representative sample pools from each fraction. All qPCR reactions were measured on a CFX384 Real-Time PCR Detection System (Bio-Rad). Statistical significance on geo-mean normalised data [64] was tested using F-test for normality and Student's t-test.

### EV characterization

Isolated EV suspensions were further characterised in terms of morphology, particle size and concentration as well as protein cargo. Calf EV samples 1 h postprandial had to be omitted since plasma yield was very low and no EV isolations could be performed. A workflow for all EV characterization related analyses is provided in Fig 2.

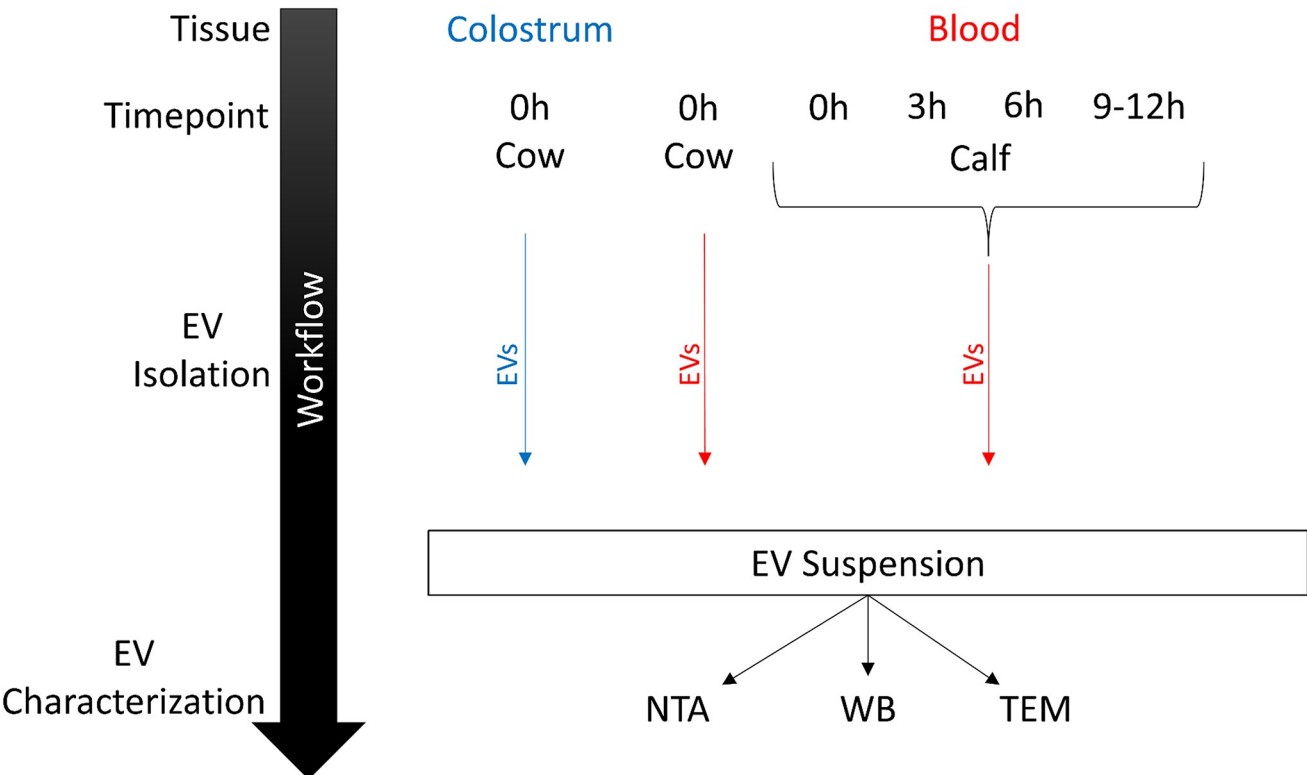

**Fig 2. Schematic overview of sample groups for EV characterization.** EVs were isolated by differential ultracentrifugation and sucrose density gradient in colostrum and blood samples from calves and cows before first feeding and at three defined postprandial time points (n = 3 each). EV morphology and protein cargo were characterized by Nanoparticle Tracking Analysis, Western blot and transmission electron microscopy.

## Nanoparticle tracking analysis

For assessment of particle concentration and size distribution, samples from 30% and 40–50% sucrose density gradient fractions were analysed using NTA. EVs were diluted in particle-free PBS and measured on a NanoSight NS300 (NTA 3.0 software, Malvern Instruments) device outfitted with a 405 nm laser and a high-sensitivity sCMOS camera. Samples were injected manually, and eight videos of 45 s each were captured under previously optimised conditions (25 frames/sec, camera level 14, FTLA algorithm). For analysis, a conservative detection threshold, auto settings for blur and minimum track length as well as a minimum of 4000 completed tracks per sample were used. Final concentrations were calculated in relation to 1 ml of plasma and skimmed colostrum.

## Transmission electron microscopy

Diluted EVs were adsorbed onto glow-discharged, carbon-coated copper grids (Quantifoil) for 2 min before manually removing excess liquid by filter paper. Grids were negatively stained in 2% uranyl acetate for 2 min and air-dried prior to imaging. All images were taken on a JEOL JEM 100CX electron microscope at 100 kV.

## Western blot

EV fractions were lysed in ice-cold detergent lysis buffer (0.1% Triton X-100 in PBS) supplemented with protease inhibitors (cOmplete Mini Protease Inhibitor Cocktail, Roche). To enhance rupture of membranes, lysates were sonicated for one minute in a water bath prior to protein quantification using BCA assay (Sigma Aldrich). For SDS-PAGE, samples were heated for 5 min at 70 ˚C in reducing Laemmli buffer transferred to 0.45 μm nitrocellulose membranes (GE Healthcare Life Sciences) using NuPAGE transfer buffer (Invitrogen) supplemented with 10% methanol. Post transfer, membranes were blocked with 1% skim milk powder in PBST for 1 h at room temperature and incubated with primary antibodies at 4 ˚C overnight. After three washes with blocking buffer, HRP-conjugated secondary antibodies were applied to membranes for 1 h at 4 ˚C. Blots were developed using Luminata Classico Western HRP substrate (Merck KGaA). Primary antibodies were purchased from Santa Cruz (goat anti-CD63, sc-31214, 1:1000; goat anti-BTN1A1, sc-324834, 1:1000), Sigma-Aldrich (rabbit anti-MFGE8, HPA002807, 1:2000), Abcam (mouse anti-HSP70, ab2787, 1:800) and Biomol (goat anti-Calnexin, WA-AF1179a, 1:2500). Secondary antibodies were from Abcam (goat anti-rabbit HRP, ab97080, 1:6700; rabbit anti-goat HRP, ab97105, 1:6700).

# Results

## Blood and colostrum from adult cows bear two distinct EV populations

Particles with characteristical EV morphology and size were found in blood as well as colostrum EV preparations by TEM (S1 Fig). No apparent size differences between tissues or sucrose density fractions could be detected and the majority of vesicles were less than 150 nm in diameter. Subsequent analysis by NTA confirmed these findings with mean and mode vesicle diameters ranging from 129.6 nm and 98.3 nm in 30% SDG colostrum EV samples to 167.8 nm and 112.4 nm in 40–50% SDG blood EV samples, respectively (Fig 3A). Although particles showed little size heterogeneity, striking differences were found for particle concentrations. EV preparations from colostrum showed very high mean particle numbers per ml (4.59E10 ± 4.25E9 P/ml) and consisted predominantly of particles floating in 40–50% SDG with only a very minor amount of particles originating from the 30% SDG fraction ($p \leq 0.05$, Fig 3B). Contrary to that, total particle numbers in cow blood were reduced by over 10-fold

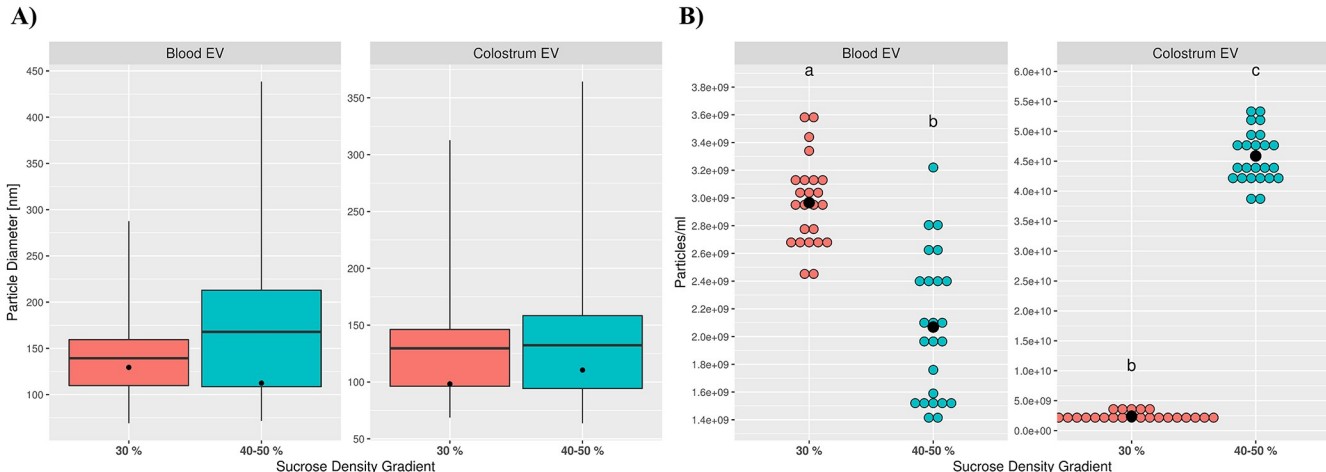

**Fig 3. Analysis of particle size and concentration by NTA in adult cows.** No significant differences in particle size were detected in blood- and colostrum-derived cow EV suspensions (A). Whiskers indicate 1st and 99th percentiles; line: mean diameter; dot: modal diameter. Blood- and colostrum-derived cow EV suspensions showed diverging concentrations of particles from different flotation densities of SDG (B). Mean particle numbers per ml of colostrum or plasma are depicted as black dots; binned coloured dots indicate individual measurements of three animals, each repeated eight times; differing letters above dots indicate significant differences in particle numbers (p<0.05).

(2.97E9 ± 3.15E8 P/ml) and were significantly enriched in 30% SDG particles (p ≤ 0.05). Additional information on particle size and concentration measurements from NTA can be found in S1 Table.

To further differentiate between blood and colostrum particles, EV- and milk-specific proteins as well as a negative marker were assessed by Western blot for 40–50% SDG preparations (Fig 4). Particles from both fluids were positive for CD63 and HSP70, two commonly used markers for vesicles, with EVs isolated from colostrum showing higher intensities, while no signals could be detected for calnexin, a marker for non-vesicular membrane contamination. A different pattern was observed for MFGE8 and BTN1A1, both of which were previously found to be highly associated with milk vesicles [65,66]. Colostrum EV isolates displayed very strong signals, whereas protein lysates from blood EV preparations were negative for these markers, which we therefore considered specific to milk EVs.

## Calf plasma levels of EVs that share colostrum characteristics are increased after feeding

Comparable to EV isolates from adult cows, particles in EV preparations from calf blood showed similar size ranges with mean and mode diameters ranging from 142.3 nm and 122.1 nm to 160.3 nm and 142.9 nm, respectively, and no significant size differences between sampling time points or SDG fractions (Fig 5A). Particle concentrations for 30% SDG were stable over all time points with no significant changes in particle numbers (Fig 5B). Meanwhile, concentrations of vesicles isolated from 40–50% SDG were consistently and significantly (p ≤ 0.05) increased for every time point compared to pre-feeding samples, reaching a maximum concentration of 1.97E10 ± 7.33E9 P/ml at 9–12 h postprandial (p ≤ 0.05) and were significantly more abundant compared to 30% SDG particles. Further similarities between EVs isolated from colostrum and calf plasma after feeding were found by Western blot analysis. CD63 and the colostrum-specific BTN1A1 were detected with increasing signal intensities with progressing time points, albeit with no or only weak expression for 0 h samples.

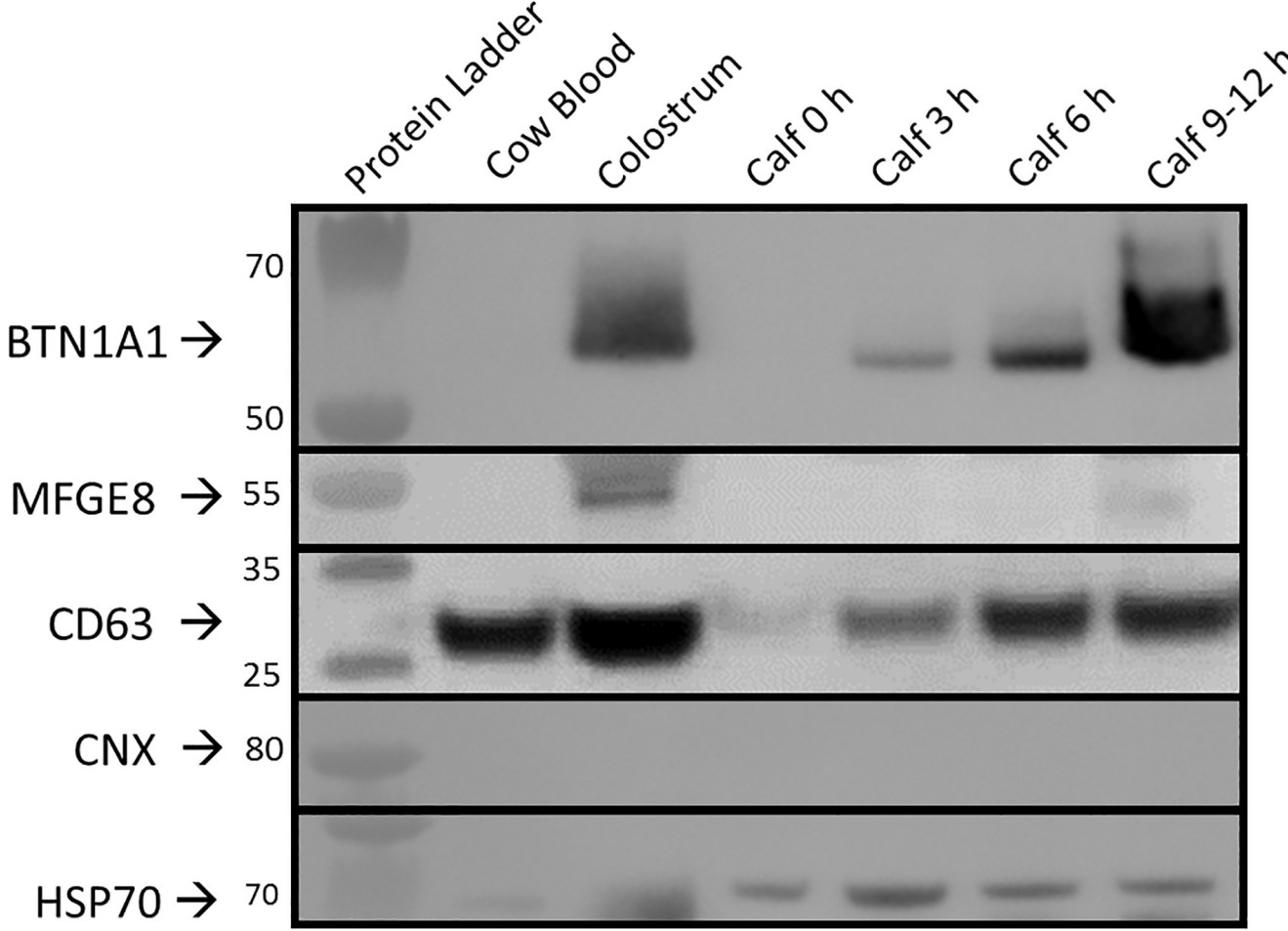

**Fig 4. Protein expression analysis of EVs of colostral and blood EVs from adult cows and calves by Western blot.** Milk specific markers BTN1A1 and MFGE8 were found in colostral and postprandial calf EV isolations but not in cow blood EVs or calf EVS pre-feeding. Presence of EV markers was confirmed by transmembrane protein CD63 and cytosolic protein HSP70 expression in all sample groups while CNX, a marker for contamination with cellular fragments, was negative. Results are representative for three biological replicates for both cow and calf samples as well as postprandial time points.

Postprandial samples were positive for MFGE8 as well, but for no more than a single time point in each calf and only after 6 or 9–12 h (Fig 4).

## Analysis of tissue- and compartment-specific small RNA profiles by NGS

Mean library sizes generated by small RNA sequencing differed within expected dimensions, ranging from 5.01E6 ± 1.30E6 in calf EV samples, potentially reflecting low input amounts for library preparation due to low RNA yield, to 1.34E7 ± 2.17E6 in colostrum cells. One sample each from whole colostrum, colostrum cells as wells as calf EV 0 h groups had to be excluded from further analysis, since they failed to amplify correctly during sequencing. Differences in small RNA profiles were assessed by aligning reads against miRNA and isomiR sequences as well as other major small RNA classes (rRNA, tRNA, snRNA, snoRNA) and plotting them together with unmapped and short reads (<16 nt) as percentages of total library size per group (Fig 6). Highest enrichment for isomiRs including canonical miRNAs was seen in whole blood samples from cow and calf along with calf blood cells (>80%) with little to no other small RNA species present. On the other hand, libraries from calf EVs were dominated by a large

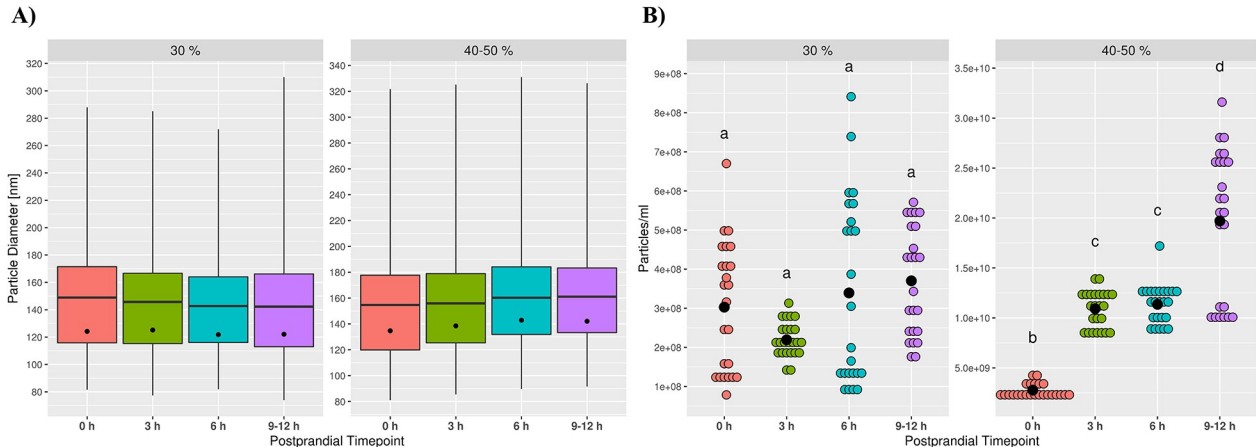

**Fig 5. Analysis of particle size and concentration by NTA in calves.** No significant changes in particle size were detected in calf EV suspensions by NTA analysis (A). Whiskers indicate 1st and 99th percentiles; line: mean diameter; dot: modal diameter. Particle numbers were significantly increased in progressing postprandial time points for EV isolates from higher density SDG fractions (40–50%), while concentrations from 30% SDG EV suspensions remained stable (B). Mean particle numbers per ml of plasma are given as black dots; binned coloured dots indicate individual measurements of three animals, each repeated eight times; differing letters above dots indicate significant changes in particle numbers.

number of unmapped sequences not belonging to any of the major small RNA classes, and a comparatively low number of miRNAs. Similar to blood, colostrum-derived samples exhibited a clear distinction between extracellular, EV-associated samples and unfractionated or cellular groups, respectively. All colostrum samples displayed higher relative numbers of tRNA reads compared to blood samples, which was most pronounced in colostrum cells. Unfractionated colostrum and colostrum EVs, on the other hand, contained the highest frequencies of short reads. Additionally, a strong relative enrichment of miRNA reads was observed in colostrum EVs compared to colostral cellular samples. Further information on alignment distributions and library sizes can be found in S2 Table.

## Analysis of canonical miRNAs reveals an influence of colostrum feeding on expression profiles in calf EVs only

Differential regulation of canonical miRNAs between colostrum and blood compartments along with postprandial time points within the same sampling group was assessed using DESeq2 and applying conservative filtering criteria (adjusted p-value $\leq 0.05$, $\log_2$ fold changes $\geq |1|$, baseMean $\geq 50$). Unfractionated colostrum differed minimally from colostrum EVs with only 9 significantly regulated miRNAs, while both groups showed a considerably different miRNA expression profile compared to colostrum cells (Table 1), reflecting RNA species distribution seen during alignment. The biggest expression changes in the data set could be found between all three colostrum-derived sample groups and unfractionated cow blood with over 150 differentially expressed miRNAs in each group (Table 1). Albeit expression profiles of calf blood-derived samples displayed high diversity between individual compartments with numbers of significantly regulated miRNAs ranging from 105 to 147 (Table 1), expression changes within different postprandial time points of blood compartments were scarce with the exception of calf EVs. Similar to calf blood cell samples, which exhibited no alterations at all over all time points, miRNA expression in unfractionated blood samples was very stable with a total of two significant regulations after 9–12 h (Table 2). On the other hand, expression changes in postprandial calf EVs samples compared to pre-feeding samples ranged from two (after 1 h) to 24 (after 9–12 h) miRNAs and continually increased in magnitude with progressing time

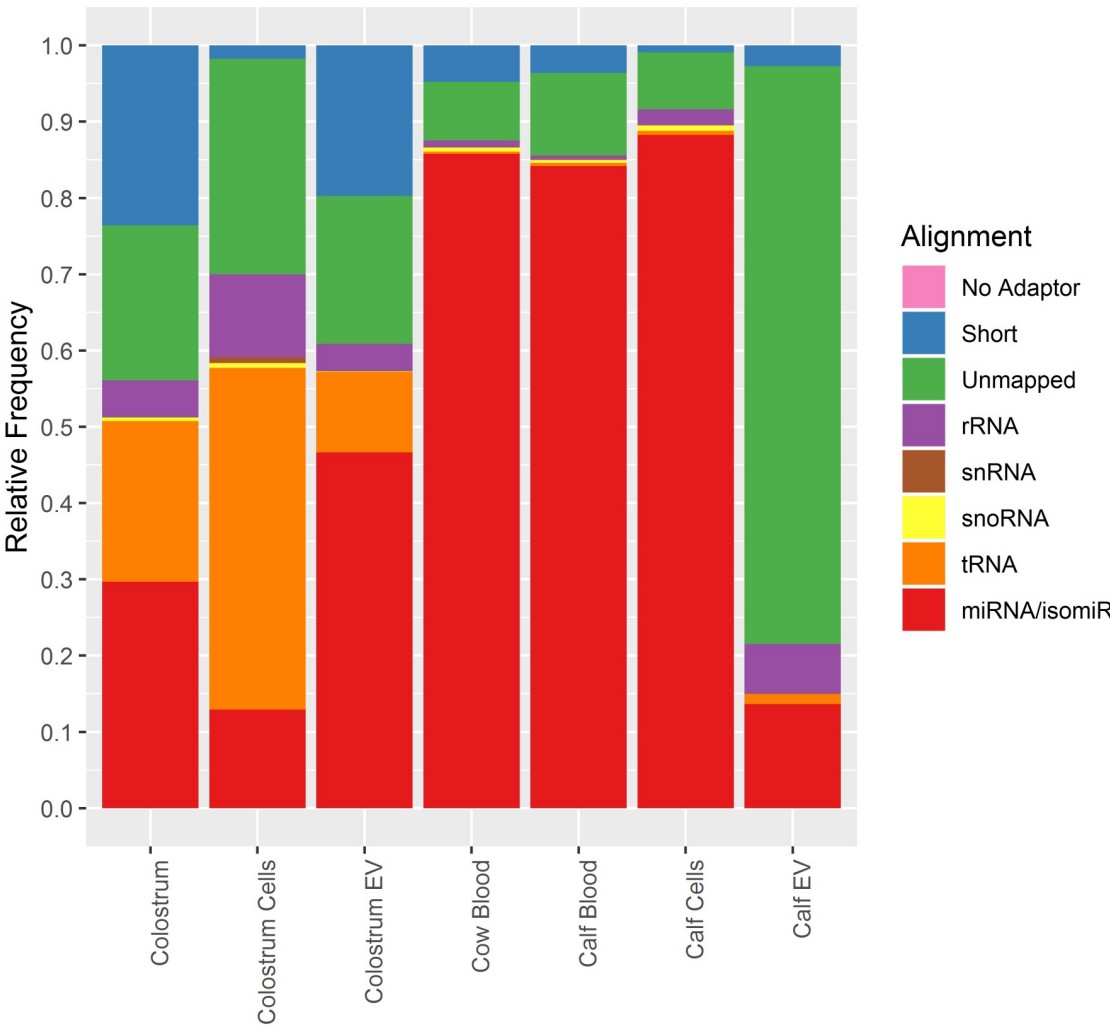

**Fig 6. Distribution statistics of analysed small non-coding RNA (ncRNA) species in relation to total library sizes of raw reads.** Distinctive RNA profiles of small ncRNA are recognizable for samples from different body fluids (colostrum, blood) as well as between unfractionated, cellular and EV-related samples. No Adaptor: reads without detectable adaptor sequence at 5'-end; Short: reads shorter than 16 nt; Unmapped: reads not mapping to either rRNA, tRNA, snRNA, snoRNA or miRNA/isomiRs sequences. Relative alignment frequencies are given as mean percentages of total library sizes for each sample group.

**Table 1. Numbers of canonical miRNAs differentially regulated between unfractionated, cellular and EV-related sampling groups in colostrum and blood.**

|  | Colostrum EVs | Colostrum | Colostrum Cells | Cow Blood |
|---|---|---|---|---|
| **Colostrum EVs** |  |  |  |  |
| **Colostrum** | 9 |  |  |  |
| **Colostrum Cells** | 92 | 69 |  |  |
| **Cow Blood** | 167 | 165 | 158 |  |
|  | **Calf Blood EVs** | **Calf Blood** | **Calf Blood Cells** |  |
| **Calf Blood EVs** |  |  |  |  |
| **Calf Blood** | 147 |  |  |  |
| **Calf Blood Cells** | 122 | 105 |  |  |

**Table 2. Numbers of differentially regulated canonical miRNAs between postprandial time points and pre-feeding samples (0 h) for calf blood-derived samples.**

|  | Postprandial Time Points | | | |
|---|---|---|---|---|
|  | **1 h** | **3 h** | **6 h** | **9–12 h** |
| **Calf Blood** | 0 | 0 | 0 | 2 |
| **Calf Blood Cells** | 0 | 0 | 0 | 0 |
| **Calf Blood EVs** | 6 | 15 | 24 | 26 |

(Table 2). Although 28 out of the total 30 canonical miRNAs with significant postprandial expression changes were up regulated, 12 of those miRNAs showed significantly smaller abundances in colostral EV samples compared to calf EV 0 h samples. Furthermore, out of the top 15 most highly expressed miRNAs in colostral EVs only three showed a significant increase after feeding with all of them belonging to the canonical miR-200a/b/c family. Differentially expressed miRNAs together with corresponding $\log_2$ fold changes of key comparisons as well as raw read counts are provided in S3 and S4 Tables.

Hierarchical clustering analysis of total miRNA expression confirmed expression profile changes detected in DESeq2 results by clearly separating unfractionated from cellular and EV-associated sample groups (Fig 7). Gene expression differences within unfractionated calf blood, blood cells and, to a smaller degree, blood EVs displayed remarkable homogeneity, resulting in groups defined by individual animals rather than clusters of particular postprandial time points. Furthermore, miRNA expression in calf blood EVs resembled colostrum profiles much more closely than any other blood-derived sample from cow or calf. Continuing the pattern revealed in RNA species distribution and differential gene expression, colostrum cells could be clearly separated from unfractionated colostrum and colostrum EVs. The potential physiological impact of up-regulated miRNAs in postprandial calf EVs was evaluated by enrichment analysis of KEGG pathways as proposed by Godard and van Eyl [62]. The most highly enriched pathway across all time points was insulin signaling, followed by TGF-beta

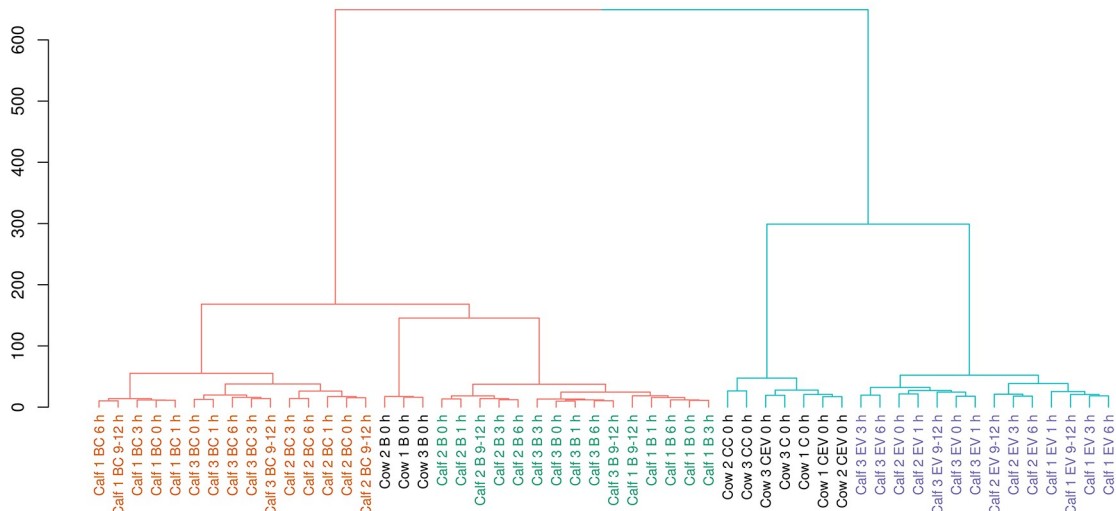

**Fig 7. Hierarchical clustering analysis of canonical miRNAs across all sample groups and postprandial time points.** The two dominant clusters were composed of colostral and calf EV samples (right) and unfractionated blood and blood cells (left). B = unfractionated blood; BC = blood cells; EV = blood extracellular vesicles; C = unfractionated colostrum, CC = colostral cells; CEV = colostral extracellular vesicles.

signaling and cytokine-cytokine receptor interaction, but no apparent overall regulation pattern could be discerned. Top 20 enriched pathways along with involved miRNAs can be found in S5 Table.

### isomiR expression profiles in postprandial calf EVs suggest non-colostral origin of up-regulated miRNA isoforms

To assess whether miRNA regulation changes in calf EVs could be attributed to colostral EV cargo, the distribution of miRNA isoforms was analysed. Since isomiR expression is highly specific for tissues and developmental stages [7–9], it allows for more precise clustering and higher confidence in determining the possible origin of expression changes in transfer studies. Unsupervised clustering of the 500 highest variance isomiRs via principal component analysis (PCA) highlighted uniformity of expression in colostral and non-EV blood samples and revealed an underlying expression pattern correlating nicely with advancement of postprandial time points (Fig 8). Although PCA uncovered a progressing overlap in expression between calf EV and colostrum-derived samples, evident on the first principal component (PC1), calf EV samples were even stronger characterised by an isomiR expression distinctively different from colostrum (PC2). Furthermore, the majority of postprandially up-regulated isomiRs showed limited overlap with isomiRs highly expressed in colostrum EVs (Fig 9). Out of the top 100 most abundant isomiRs in colostrum EVs, constituting 74.5% of all isomiR reads in these samples, only 30 could be rediscovered in postprandial calf EVs as significantly up regulated. Similar to canonical miRNA analysis, 15 of these isomiRs belonged to the same miRNA family of bta-miR-200a/b/c.

## Discussion

The role of dietary miRNAs and their transfer to consumers has recently provoked a lot of discussion in the scientific community. Although modes of transfer, uptake and distribution to recipient cells are still not clearly deciphered, the possible ramifications on our understanding of environmental influences on gene expression are colossal due to the essential role of miRNAs in post-transcriptional gene regulation. The debate is further fueled by the association of many dietary miRNAs with EVs, providing not only a robust carrier to resist digestion but also a conceivable mean of transport to the gastrointestinal tract [25–29]. In this study, we assessed the potential transfer of colostral EVs and their protein and miRNA cargo based on biochemical, molecular analyses and high-throughput sequencing of small non-coding RNAs. As recent findings on the bioavailability of milk miRNAs are partly in direct contradiction to each other [30,31], we chose samples from blood and colostrum of multiparous cows as well as from the circulation of their direct offspring before and after their first feeding as the model with a high probability to demonstrate an uptake of dietary EVs.

The ability to isolate reasonably pure vesicles is a prerequisite for any viable EV study [67,68]. Following well-established protocols for differential ultracentrifugation in combination with flotation into a density gradient [21,43] yielded particles in the size range of typical small EVs in all biofluids. Modal diameters of particles (~98 to 112 nm) were slightly above exosome-like size (30–100 nm) [69] and in line with previous findings on EVs from bovine or human milk and blood [25,44,70] (Fig 3A). No significant differences in size could be detected between blood and colostrum samples or particles recovered from different sucrose density fractions, hinting at a high reproducibility of EV isolation methods and low inter-individual variability. Quantitative analysis of particles on the other hand, revealed prominent differences in particle composition and concentration in blood and colostrum (Fig 3B). In adult animals, EV numbers were more than 10 times higher in colostral samples compared to blood samples.

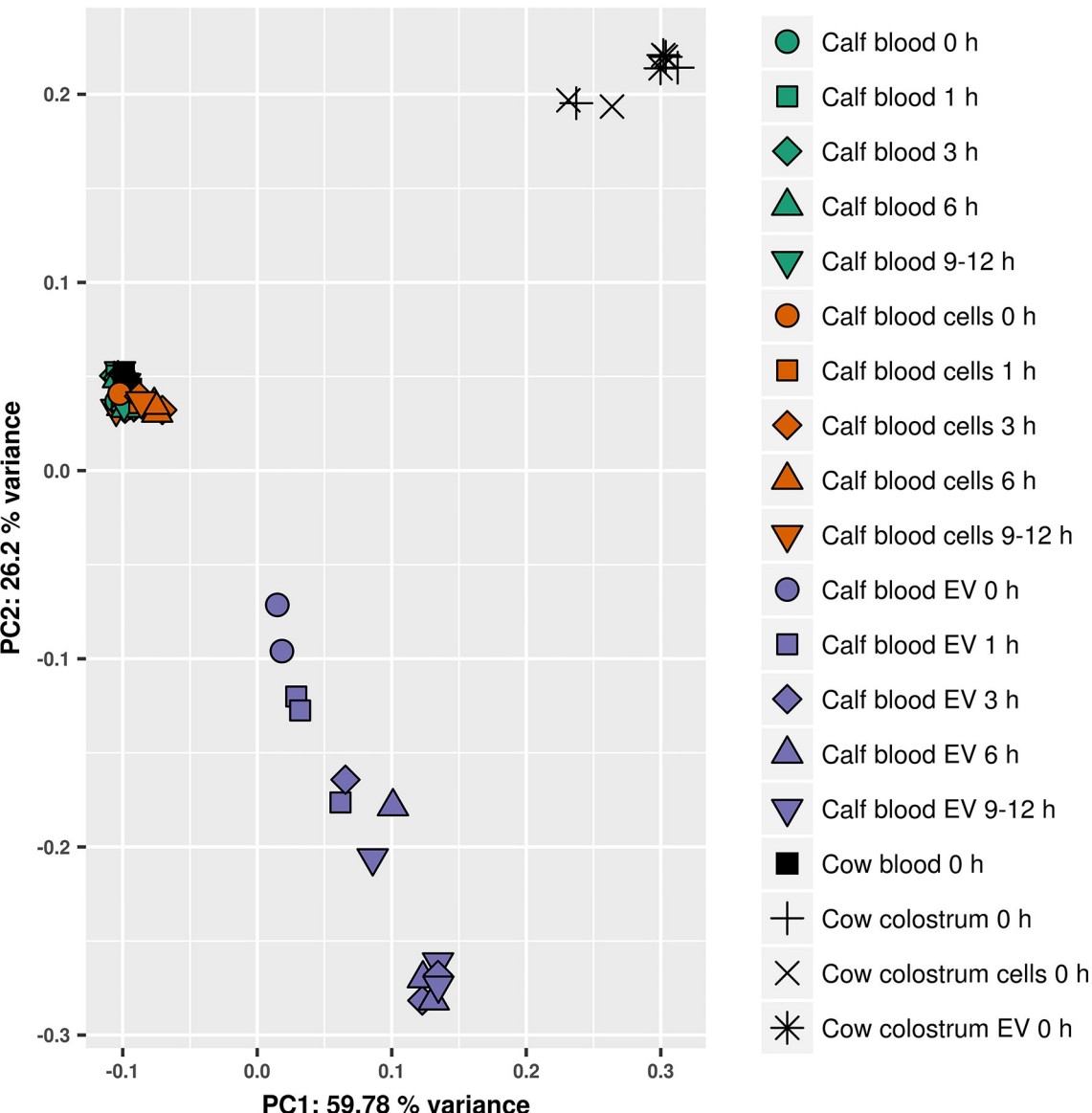

**Fig 8. Principal component analysis of the top 500 most varying isomiRs across all samples.** Calf sample groups are denoted by different colours while postprandial time points are indicated by symbol shapes. Cow samples are given in black with different symbol shapes depicting sample groups.

While this stands in contrast to the findings of Koh et al, who found concentrations of milk particles to be ~3.2 fold lower than plasma particles [70], it is corroborated by previous reports that have shown that miRNAs associated with colostrum EVs are highly enriched compared to milk from later lactation stages [20,21,24]. Furthermore, the majority of colostrum EVs were significantly denser (40–50% SDG) than blood-derived EVs (30% SDG), although all isolated particles generally fell in the density range expected for exosome-like vesicles (1.1270–1.2296 g/ml) [69]. Predominant EVs of high densities in colostrum were in agreement with findings from Hata et al., showing highest content of vesicle-associated proteins and RNAs from density fractions corresponding to 1.20 g/ml compared to lower densities [21]. While density of

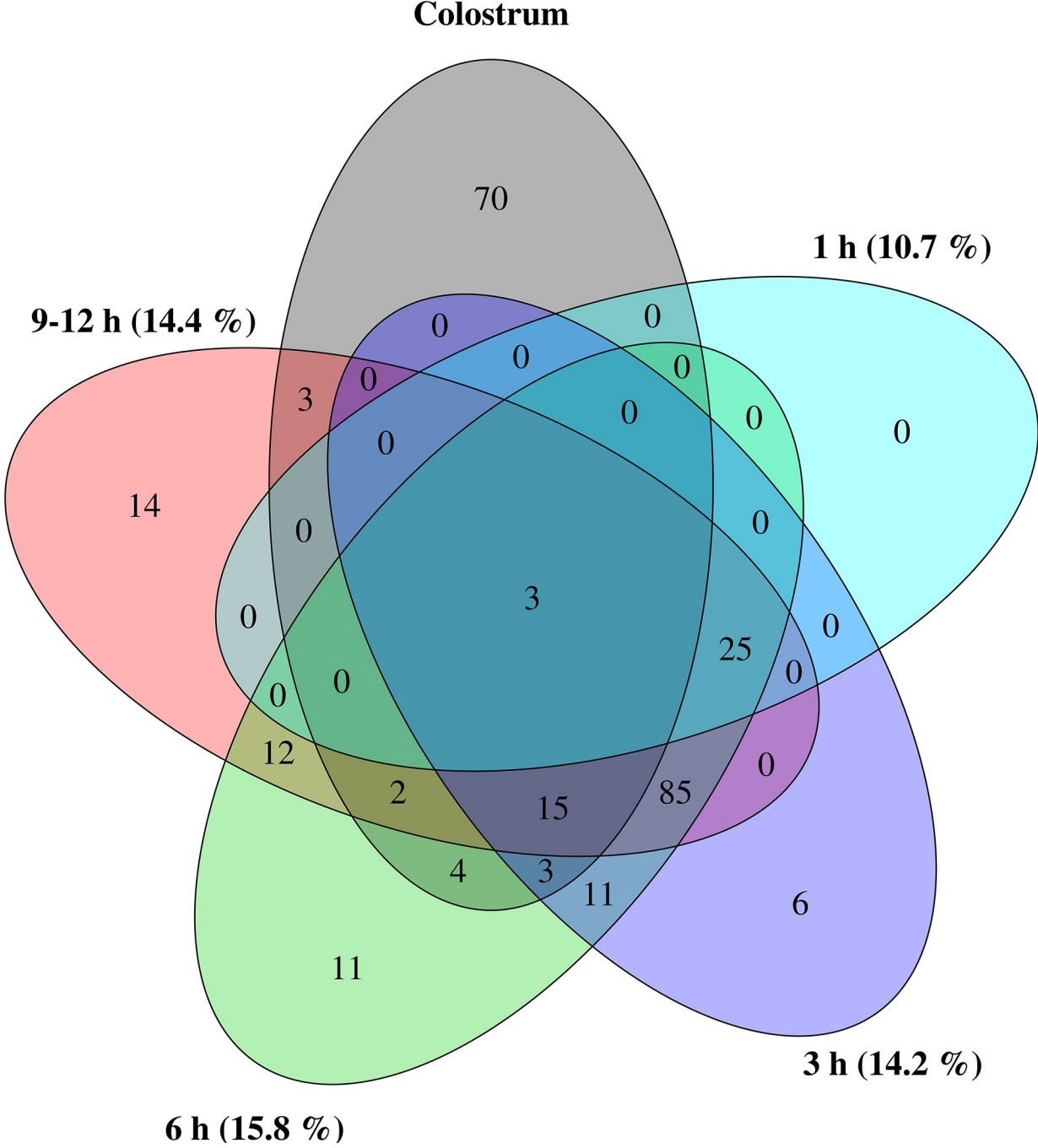

**Fig 9. Venn diagramm displaying the overlap between the top 100 most abundant isomiRs in colostrum EVs and significantly upregulated isomiRs in postprandial calf EV samples of all time points.** Number in brackets denote the percentage of common isomiRs per time point.

particles from plasma was in line with numerous reports [71,72], this is to the best of our knowledge the first time that a significant shift in density between bovine EV populations of blood and milk has been reported. It should be noted, however, that concentration measurements of particles by NTA are prone to overestimation due to co-isolated contaminants such as lipoproteins and protein aggregates, especially in vesicle preparations of low purity [73–75].

Particularly preparations from milk or colostrum, with its high content of fat globules and senescent, ex-foliated epithelial cells seem prone to accumulate unwanted particles that can mimic EV properties for example by aggregation during ultracentrifugation [76]. Nevertheless, detected EV concentrations in adult blood and colostrum seem genuine as most contaminants should be discarded in SDG fractions of higher (protein aggregates) or lower (aggregated small fat globule membranes) densities [43]. Endosomal origin of membranous EV preparations as well as higher concentrations of particles in colostrum were further supported by Western blot analysis of positive (CD63, HSP70) and negative (Calnexin) vesicle markers (Fig 4). Positive expression of BTN1A1 in colostrum-derived EVs is in concordance with recordings on Vesiclepedia and ExoCarta [77,78], the biggest databases on molecular data of extracellular vesicles, which list BTN1A1 as exclusively associated with bovine and human milk EVs [65,66]. Expression of MFGE8 however, was reported to be widespread in EVs originating from a large variety of sources including B cells and platelets [79,80]. Although MFGE8 was never reported to be associated with bovine EVs apart from milk [66] and we considered absent expression in cow blood to be genuine, it could also not be excluded that a lack of signal originated from low starting input material due to the fact that only samples from 40–50% SDG fractions were analysed in Western blots.

Differentiation between blood- and colostrum-derived samples and especially EVs was further driven by abundances of small RNA species analysed by Next-Generation Sequencing. High throughput sequencing has become the tool of choice for analysing nucleic acids due to its high precision in quantifying of single RNA sequences and its accurate detection of diverse RNA compositions. Clear differences were found in relative frequencies of miRNA and tRNA as well as reads shorter than 16 nt between whole blood and colostrum-derived samples in cows (Fig 6). Higher incidences of short reads most likely stem from an increased number of degradational products tracing back to ex-foliated, senescent epithelial cells present in colostrum, while enrichment of miRNAs in colostrum vesicles compared to the cellular fraction has been reported before by Sun et al [20]. Additionally, varying expression both between blood- and colostrum-derived samples and within cellular and extracellular fractions in colostrum were confirmed by differential analysis of canonical miRNAs (Table 1). In general, miRNA expression profiles in colostrum-derived samples were in concordance with previous reports with a substantial overlap in the most abundant miRNAs [21,81,82]. miRNA expression in blood and colostrum was completely dissimilar, duplicating results obtained from milk and peripheral blood cells [83] as well as circulating miRNAs [84]. However, differences between unfractionated colostrum and colostrum EVs were minuscule compared to colostrum cells, suggesting that the majority of miRNAs in colostrum is present in the extracellular compartment and the removal of the upper fat layer and its associated miRNAs is of minor consequence. Nonetheless, a potential impact from colostral fat derived miRNA cannot be ruled out completely and should be analysed in further studies.

Taken together, quantitative and qualitative analyses of EVs in cows indicate distinct populations of vesicles in blood and colostrum, enabling the identification of a potential postprandial transfer of colostrum EVs into the blood circulation based upon EV concentrations, protein cargo and miRNA expression. Vesicles identified in calf blood after feeding indeed shared characteristics classified as specific for colostrum EVs (Fig 5). Particle concentrations for preparations increased significantly in a time-dependent manner solely for 40–50% SDG, from levels similar to cow blood to approximately half the abundance in colostrum. Contrary to adult cow blood EVs, calf particle numbers from the 30% SDG fraction even at pre-feeding time point were significantly lower by an order of magnitude compared to high-density vesicles. Similar distributions of EVs as encountered in cow blood EVs could potentially be acquired later during development, but it is also conceivable that higher amounts of 40–50%

SDG EVs at time point 0 h are the product of autonomous but very limited feeding before first blood samples could be drawn even though all calves were under the supervision of a milker. Protein expression of high-density particles was positive for both milk-specific markers, although only BTN1A1 followed the progressing pattern of increase indicated by particle concentrations. Divergent expression of MFGE8 with only low abundances detectable after 6 or 9–12 h suggests either the existence of multiple milk EV subpopulations in 40–50% SDG fractions, with greatly differing transfer efficiency through the intestinal epithelium and implying selective mechanisms for uptake, or a breakdown of milk EVs in epithelial cells followed by repackaging of protein cargo in a selective and directive manner.

Small RNA expression profiles and in-depth analyses of isomiR distributions in calf blood-derived samples however, painted a somewhat different picture of the bioavailability of dietary EVs and their cargo from colostrum. While miRNA expression contrasted significantly between all three blood-derived samples (Table 1), postprandial changes within sample groups could only be detected for calf EVs (Table 2). Absence of miRNA expression changes in unfractionated calf blood and blood cells was also apparent from hierarchical clustering analysis with sub-clusters within sample groups being defined by inter-animal differences rather than postprandial time points (Fig 7). Even though a time-dependent influence of colostrum feeding on expression of miRNA and their isoforms in calf blood EVs was evident (Table 2 and Fig 8), ostensible similarities with colostrum EVs might be misleading. In fact, the only consistent overlap for highly expressed miRNAs between colostrum EVs and calf blood EVs from postprandial time points that could be detected, was from a single miRNA family consisting of miR-200a, miR-200b and miR-200c. Furthermore, nearly half of all significant expression changes in post-feeding EV samples were inversely regulated compared to colostrum EVs. To help differentiate between endogenously produced miRNAs and miRNAs potentially taken up from dietary sources, isomiR patterns were analysed. Although a number of miR-200 isoforms could be detected with comparable differential expression, the most prominent isomiRs from colostrum EVs were missing in postprandial calf EVs and vice versa (Fig 9). A more likely explanation for the significant increase of miR-200a/b/c, instead of an uptake from dietary sources, might be its role in the processing of food-related signals. Next to its well-documented function in tumor development and progression [85], the miR-200 family regulates key players (e.g. FOG2, Rheb) in the insulin signaling pathway [86–88]. The enrichment analysis of KEGG pathways, indicating insulin signaling as the top result, further promotes the idea that up-regulated miRNAs in calf EVs after colostrum feeding are primarily involved in the regulation of food-related energy uptake in recipient cells.

In conclusion, our findings on the bioavailability of colostrum-derived EVs in the bloodstream of neonatal calves suggest an even more complex mode of uptake than previously assumed [10,30]. The unequal uptake of protein and miRNA cargo discourages the hypothesis of para- and transcellular transport of all intact dietary EVs through the intestinal epithelium into the blood. The reason that we readily found colostrum-specific protein markers but could not detect any meaningful uptake of miRNAs could well be attributable to their respective localization in or on the EV. Postprandial expression of BTN1A1 as well as MFGE8, both membrane-associated proteins, is in line with studies on orally administered labeled exosomes that utilised a lipophilic membrane dye [33]. miRNAs, on the other hand, are thought to be incorporated into the lumen of the vesicle and, like us, a number of studies have failed to detect an uptake of dietary miRNAs so far [13,31,32]. One possible conclusion to this decoupling of membrane-associated cargo from luminal miRNAs, might be an uptake of dietary EVs into intestinal epithelial cells, followed by disassembly of the EV prior to repackaging of EV protein cargo and delivery into the blood. Evidence for this was recently

given by a small study deducted by Manca et al., who showed distinct localization of EVs (mainly liver) and their miRNA cargo (mainly brain and kidney) [89]. Another likely reasoning could be an unequal uptake of different EV subpopulations within colostrum. EVs in milk and colostrum are mainly derived by two independent secretory pathways. Firstly by the classic endosomal multivesicular body pathways common to all exosome secretion and secondly by a Golgi-endoplasmic reticulum-derived pathway shared with milk fat globule secretion [90,91]. Due to the high overlap of milk exosomal proteomes with other exosomes independent from their tissue of origin [90], we chose two proteins (BTN1A1, MFGE8) as indicators for milk specificity that while present in exosmal proteomes are highly enriched in milk fat globule membranes. It stands to reason that the dissimilar uptake of EV cargo could also stem from a disparate distribution in exosomal EVs (miRNA) and EVs derived from the Golgi-endoplasmic reticulum pathway (BTN1A1, MFGE8). Based on our experiments we cannot rule out the possibility that miRNAs from colostrum EVs were either not taken up at all, remained in the intestinal epithelium, or were directly transported to recipient tissues without extended circulation in the blood. Furthermore, we cannot exclude the possibility that our sampling points (1h, 3h, 6h and 9-12h), might have missed the point of maximum colostral EV uptake in general or exosomal uptake in particular. Further in-depth studies on kinetics of colostral EVs uptake will be of great help in defining optimal sampling time points. To discern final destinations of miRNAs, further investigations including sampling from intestinal tissues as well as likely recipient organs such as liver or kidneys are needed.

## Supporting information

**S1 Fig. Morphology of colostral and blood EVs by transmission electron microscopy.** Images are representative for three separate biological replicates per sample group. No differences were observed for postprandial time points in calf blood EV.
(TIFF)

**S1 Table. Particle size and concentrations of EV suspensions as determined by NTA analysis.** Measurements are summarized as mean values and standard deviations within a sampling group and size distributions are further characterized by giving the most frequent particle size.
(DOCX)

**S2 Table. Distribution of raw read counts on small ncRNA species and inappropriate length classes.** Short: smaller than 16 nt, No Adaptor: 50 nt long with no detectable adaptor sequence at 5'-end. Read numbers are given as mean values for each sample group.
(DOCX)

**S3 Table. log2 fold-changes of significantly regulated canonical miRNAs in colostral and postprandial calf blood EV compared to pre-feeding calf blood EVs.** Green background indicates up-regulation compared to calf EV 0h samples while red background highlights down-regulation. Expression changes without significancy are denoted by n.s.
(DOCX)

**S4 Table. Raw read counts of significantly regulated canonical miRNAs in colostral and postprandial calf blood EV compared to pre-feeding calf blood EVs.** Dam, calf pairs are denoted by (A)-(C); sample types are abbreviated as follows B = unfractionated blood, BC = blood cells, EV = extracellular vesicles, M = unfractionated colostrum, MC = colostrum cells; time points are denoted as 0h-9-12h.
(XLSX)

**S5 Table. Gene set enrichment of the top 20 KEGG pathways based on significantly up-regulated canonical miRNA in postprandial time points.** Analysis was performed on targets of human homologous miRNAs obtained from miRTarBase supported by strong experimental evidence (Reporter assay or Western blot).
(DOCX)

**S1 Raw images.**
(PDF)

## Author Contributions

**Conceptualization:** Benedikt Kirchner, Vijay Paul, Michael W. Pfaffl.

**Data curation:** Benedikt Kirchner, Dominik Buschmann, Vijay Paul.

**Formal analysis:** Benedikt Kirchner, Dominik Buschmann, Michael W. Pfaffl.

**Software:** Benedikt Kirchner, Michael W. Pfaffl.

**Validation:** Benedikt Kirchner.

**Visualization:** Benedikt Kirchner.

**Writing – original draft:** Benedikt Kirchner.

**Writing – review & editing:** Dominik Buschmann, Michael W. Pfaffl.

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
