## [Decision Letter · Decision Letter 0]

10 Oct 2019

PONE-D-19-23560

Postprandial transfer of colostral extracellular vesicles and their protein and miRNA cargo in neonatal calves

PLOS ONE

Dear Mr. Kirchner,

Thank you for submitting your manuscript to PLOS ONE. After careful consideration, we feel that it has merit but does not fully meet PLOS ONE’s publication criteria as it currently stands. Therefore, we invite you to submit a revised version of the manuscript that addresses the points raised during the review process.

Important points are : 1) the acknowledgments of the experimental limitations of the study; 2) the possible bias resulting from overlooking the microRNAs contained in the upper fat layer of the colostrum; 3) providing in as much as possible absolute copy numbers of microRNAs. 

We would appreciate receiving your revised manuscript by Nov 24 2019 11:59PM. To enhance the reproducibility of your results, we recommend that if applicable you deposit your laboratory protocols in protocols.io, where a protocol can be assigned its own identifier (DOI) such that it can be cited independently in the future. For instructions see: http://journals.plos.org/plosone/s/submission-guidelines#loc-laboratory-protocols

We look forward to receiving your revised manuscript.

Kind regards,

Pierre Busson, MD, PhD, Res Director

Academic Editor

PLOS ONE

Journal Requirements:

2. In your Methods section, please include a comment about the state of the animals following this research. Were they euthanized or housed for use in further research? If any animals were sacrificed by the authors, please include the method of euthanasia and describe any efforts that were undertaken to reduce animal suffering.

3. We note that you are reporting an analysis of a microarray, next-generation sequencing, or deep sequencing data set. PLOS requires that authors comply with field-specific standards for preparation, recording, and deposition of data in repositories appropriate to their field. Please upload these data to a stable, public repository (such as ArrayExpress, Gene Expression Omnibus (GEO), DNA Data Bank of Japan (DDBJ), NCBI GenBank, NCBI Sequence Read Archive, or EMBL Nucleotide Sequence Database (ENA)). In your revised cover letter, please provide the relevant accession numbers that may be used to access these data. For a full list of recommended repositories, see http://journals.plos.org/plosone/s/data-availability#loc-omics or http://journals.plos.org/plosone/s/data-availability#loc-sequencing.

Additional Editor Comments (if provided):

Please, add page and line numbers

As emphasized by the referee, this study is susceptible to have a strong impact in a highly controversial field. Therefore, it is necessary to point some limitations of the experimental approach, for example the fact that colostral exosomes differ substantiality from mature milk exosomes or the fact that we have almost no data on the kinetics of colostral exosome uptake.

As suggested by the referee, it would be good to precise in the Materials and Methods section that the upper fat layer of the colostrum is rich in microRNAs. This fraction is not taken in account in Fig.7 although it may contribute to the profile of miRNAs in calf plasma and even calf plasma EVs.

The referee also rightly suggests providing in as much as possible absolute copy numbers of microRNAs instead of relative abundance.

The biochemical characterization of the exosomes is based on the detection of CD63 and the absence of calnexine. As proposed by the refere, it would be nice do I have additional markers like CD9 and CD81.

The legend of figure 5 is rather obscure. If I am not mistaken, there is no colostrum sample in this figure although the legend mentions “colostrum or plasma” (line 6).

One minor point. Page 5, line 15 : I wonder whether one should read “solely for 40-50% SDG” instead of “solely from…”.

Reviewers' comments:

Reviewer's Responses to Questions

**Comments to the Author**

1. Is the manuscript technically sound, and do the data support the conclusions?

Reviewer #1: No

2. Has the statistical analysis been performed appropriately and rigorously? 

Reviewer #1: No

3. Have the authors made all data underlying the findings in their manuscript fully available?

Reviewer #1: No

4. Is the manuscript presented in an intelligible fashion and written in standard English?

Reviewer #1: Yes

5. Review Comments to the Author

Reviewer #1: Review manuscript PONE-D-19-23560

Postprandial transfer of colostral extracellular vesicles and their protein and miRNA cargo in neonatal calves

Benedikt Kirchner, Dominik Buschman, Vijay Paul, Michael W. Pfaffl

General comment

The uptake of bovine milk exosomes and their miR cargo into the systemic circulation of human milk consumers is still a matter of debate in dairy research and human medicine.

With the intention to bring more light into milk-mediated systemic miR communication, the authors used bovine colostrum as a potential model for maternal-neonatal miR transfer.

To elucidate the role of bovine colostrum extracellular vesicles (EVs) and exosomes and their RNA cargo, the investigators assessed the potential postprandial transfer of colostral EVs to the circulation of newborn calves by analysing colostrum-specific proteins and miRNAs, including specific isoforms (isomiRs) in cells, EV isolations and unfractionated samples from blood and colostrum. Postprandial blood samples of calves show a time-dependent

increase in EVs that share morphological and protein characteristics of colostral EVs. Surprisingly, analysis of miRNA expression profiles by Next-Generation Sequencing

gave a different picture. Although significant postprandial expression changes

could only be detected for calf EV samples, expression profiles show very limited

overlap with highly expressed miRNAs in colostral EVs or colostrum in general. The authors concluded that from their experimental data that the uptake of membrane-associated protein cargo but not luminal miRNAs from colostral EVs into the circulation of neonatal calves may be selective.

The reviewer is not convinced that the presented colostrum-EV-based conclusions apply to transfer situation of exosomes and their miR content of mature bovine milk in humans and non-ruminant animal models. It should be emphasized that colostrum is a very unique mammary gland-derived secretory product.

Milk EVs derive from two independent secretory pathways: (1) the Golgi-endoplasmic reticulum-derived pathway common to milk fat globule secretion and (2) the endosomal multivesicular body pathway of exosome secretion (Reinhardt TA, Lippolis JD, Nonnecke BJ, Sacco RE. Bovine milk exosome proteome. J Proteomics. 2012;75(5):1486-92; Benmoussa A, Gotti C, Bourassa S, Gilbert C, Provost P. Identification of protein markers for extracellular vesicle (EV) subsets in cow's milk. J Proteomics. 2019;192:78-88).

The mean diameter of EV particles in the range of 130-160 nm clearly indicates that the investigators primarily followed the take up of non-exosome EVs, because purified bovine and equine milk exosomes have reported diameters of 50-100 nm and 30-100 nm (Reinhardt TA, Lippolis JD, Nonnecke BJ, Sacco RE. Bovine milk exosome proteome. J Proteomics. 2012;75(5):1486-92; Sedykh SE, Purvinish LV, Monogarov AS, Burkova EE, Grigor'eva AE, Bulgakov DV, Dmitrenok PS, Vlassov VV, Ryabchikova EI, Nevinsky GA. Purified horse milk exosomes contain an unpredictable small number of major proteins. Biochim Open. 2017;4:61-72). Moreover, the postprandial increase of BTN1A1 (butyrophilin) and lactadherin (MFGE8), predominant proteins of the MFG membrane (Ogg SL, Weldon AK, Dobbie L, Smith AJ, Mather IH. Expression of butyrophilin (Btn1a1) in lactating mammary gland is essential for the regulated secretion of milk-lipid droplets. Proc Natl Acad Sci U S A. 2004;101(27):10084-9. Robenek H, Hofnagel O, Buers I, Lorkowski S, Schnoor M, Robenek MJ, Heid H, Troyer D, Severs NJ. Butyrophilin controls milk fat globule secretion. Proc Natl Acad Sci U S A. 2006;103(27):10385-10390; Reinhardt TA, Lippolis JD, Nonnecke BJ, Sacco RE. Bovine milk exosome proteome. J Proteomics. 2012;75(5):1486-92), underline the preferred uptake of MFG-related non-exosomal EVs. Compared with the MFG membrane proteome, a 15–30-fold reduction in the abundance of xanthine oxidase, butyrophilin, lactadherin/MGFE8 and adipophilin/perilipin-2 has been observed in milk exosome membranes. (Reinhardt TA, Lippolis JD, Nonnecke BJ, Sacco RE. Bovine milk exosome proteome. J Proteomics. 2012;75(5):1486-92).

The concomitant presence of the three tetraspanins CD9, CD63 and CD81 was suggested to characterize an EV subset as exosomes (Kowal J, Arras G, Colombo M, Jouve M, Morath JP, Primdal-Bengtson B, Dingli F, Loew D, Tkach M, Théry C. Proteomic comparison defines novel markers to characterize heterogeneous populations of extracellular vesicle subtypes. Proc Natl Acad Sci U S A. 2016;113(8):E968-77). Furthermore, it has been recommended to determine TSG101 as an exosomal marker protein of cow´s milk (Koh YQ, Peiris HN, Vaswani K, Meier S, Burke CR, Macdonald KA, Roche JR, Almughlliq F, Arachchige BJ, Reed S, Mitchell MD. Characterization of exosomes from body fluids of dairy cows. J Anim Sci. 2017;95(9):3893-3904; Benmoussa A, Gotti C, Bourassa S, Gilbert C, Provost P. Identification of protein markers for extracellular vesicle (EV) subsets in cow's milk. J Proteomics. 2019;192:78-88). A deficit of their study is the fact that the authors only used CD63 as a potential exosome marker.

However, most critical is the collection of colostrum at the day of parturition. It has been demonstrated that there is a dramatic postnatal change of the colostrum exosome proteome compared to mature milk exosome proteome. It was observed that certain proteins that were highly abundant in 24 h colostrum exosomes, slowly diminished at 48 and 72 h and were at similar amounts to the mature milk exosome sample (Samuel M, Chisanga D, Liem M, Keerthikumar S, Anand S, Ang CS, Adda CG, Versteegen E, Jois M, Mathivanan S. Bovine milk-derived exosomes from colostrum are enriched with proteins implicated in immune response and growth. Sci Rep. 2017;7(1):5933). Early colostral exosomes are enriched with proteins implicated in immune response and intestinal cell proliferation and may not be comparable to mature milk exosomes in terms of composition, RNA content and physiological functions. Immediate postnatal exosomes are thus unique and not a representative model for mature human milk exosomes and their potential systemic beneficial and adverse effects in humans (Zempleni J, Aguilar-Lozano A, Sadri M, Sukreet S, Manca S, Wu D, Zhou F, Mutai E. Biological Activities of Extracellular Vesicles and Their Cargos from Bovine and Human Milk in Humans and Implications for Infants. J Nutr. 2017;147(1):3-10; Melnik BC, Schmitz G. Exosomes of pasteurized milk: potential pathogens of Western diseases. J Transl Med. 2019;17(1):3). Thus, the presented model using early colostrum exosomes is not transferable to the in vivo situation of exosome biology and exosome trafficking of exosomes of midlactation dairy cows. These limitations have to be outlined in the Section Limitations of this study. They are of crucial importance because the unexperienced reader may draw misleading conclusions for the in vivo exosome traffic of pasteurized commercial milk to the human milk consumer.

Another concern regarding the study design is the fact that the authors provided no kinetic studies of exosome or miR uptake. Baier et al. (see ref. 30) demonstrated that the maximum of miR-29b uptake in human volunteers 6 hours after oral intake of commercial milk. The calf is a ruminant and the reviewer is not aware of available data on colostrum and milk exosome kinetics in this species. It is not known which part of the complex bovine intestine is the major area for exosome uptake. Thus, the maximum of colostral exosome uptake and miR appearance in the calf blood plasma is an uncertainty that should be emphasized.

Specific comments

Introduction

3rd paragraph

The data of Howard et al. demonstrate that the majority of milk miR-29b (nearly two thirds) survive pasteurization, homogenization and cooled storage (see ref. 22). This is in agreement with the data reported for pasteurized cow´s milk by Golan-Gerstl et al. (Golan-Gerstl R, Elbaum Shiff Y, Moshayoff V, Schecter D, Leshkowitz D, Reif S. Characterization and biological function of milk-derived miRNAs. Mol Nutr Food Res. 2017;61(10):1700009).

The study of Baier et al. (see ref. 30) is convincing for the reviewer and other researchers. The increase in miR-29b in blood monocytes 6 h after milk consumption can be plausibly explained by exosomal miR uptake. The Auerbach et al. study (see ref. 31) had a severe technical problem as exosome samples were not continuously cooled. The authors reported that the sample arrived with largely sublimated dry ice. Thus, exosome or exosome protein degradation could not be excluded.

The cited study of Title et al (see. ref. 32) was not a well-designed study and has been criticized by various independent researchers (Melnik BC, Kakulas F, Geddes DT, Hartmann PE, John SM, Carrera-Bastos P, Cordain L, Schmitz G. Milk miRNAs: simple nutrients or systemic functional regulators? Nutr Metab (Lond). 2016;13:42).

The authors should note that bovine milk exosomes reached distant organs in independent studies in mice (Arntz OJ, Pieters BC, Oliveira MC, Broeren MG, Bennink MB, de Vries M, van Lent PL, Koenders MI, van den Berg WB, van der Kraan PM, van de Loo FA. Oral administration of bovine milk derived extracellular vesicles attenuates arthritis in two mouse models. Mol Nutr Food Res. 2015;59(9):1701-12; Manca S, Upadhyaya B, Mutai E, Desaulniers AT, Cederberg RA, White BR, Zempleni J. Milk exosomes are bioavailable and distinct microRNA cargos have unique tissue distribution patterns. Sci Rep. 2018;8(1):11321)

In contrast to the presented skepticism concerning systemic milk exosome traffic, pharmacologists regard milk exosomes as ideal carriers for systemic drug transfer because milk exosomes exhibit cross-species tolerance with no adverse immune and inflammatory response and reach distant target tissue (Munagala R, Aqil F, Jeyabalan J, Gupta RC. Bovine milk-derived exosomes for drug delivery. Cancer Lett. 2016;371(1):48-61; Betker JL, Angle BM, Graner MW, Anchordoquy TJ. The Potential of Exosomes From Cow Milk for Oral Delivery. J Pharm Sci. 2019;108(4):1496-1505). Notably, orally delivered paclitaxel (PAC)-loaded exosomes (ExoPAC) showed significant tumor growth inhibition against human lung tumor xenografts in nude mice, demonstrating a systemic effect of milk exosomes (Agrawal AK, Aqil F, Jeyabalan J, Spencer WA, Beck J, Gachuki BW, Alhakeem SS, Oben K, Munagala R, Bondada S, Gupta RC. Milk-derived exosomes for oral delivery of paclitaxel. Nanomedicine. 2017;13(5):1627-1636). Pharmaceutical industry such as Roche would not invest $ 1 Billion for milk exosome preparations for systemic drug delivery (https://www.genengnews.com/topics/omics/got-milk-roche-to-apply-puretech-exosomes-platform-in-1b-collaboration/). Thus, overwhelming evidence supports the systemic traffic and long-distance effects of bovine exosomes derived from commercial milk in humans and non-ruminant animal models.

Therefore, the reviewer disagrees with the authors´ statement that “uptake of milk-derived EVs is more likely to be observed between mother and direct offspring as opposed to a cross-species transfer to adult individuals”.

In contrast to Witwer et al. (see ref. 36), the majority of independent international milk exosome researchers (USA, Canada, Australia, India and Germany) accept the bioavailability and systemic effects of bovine milk exosomes (Manca S, Upadhyaya B, Mutai E, Desaulniers AT, Cederberg RA, White BR, Zempleni J. Milk exosomes are bioavailable and distinct microRNA cargos have unique tissue distribution patterns. Sci Rep. 2018;8(1):11321; Wang L, Sadri M, Giraud D, Zempleni J. RNase H2-Dependent Polymerase Chain Reaction and Elimination of Confounders in Sample Collection, Storage, and Analysis Strengthen Evidence That microRNAs in Bovine Milk Are Bioavailable in Humans. J Nutr. 2018;148(1):153-159. Zempleni J, Sukreet S, Zhou F, Wu D, Mutai E. Milk-derived exosomes and metabolic regulation. Annu Rev Anim Biosci. 2019; 7:245-62; Benmoussa A, Provost P. Milk microRNAs in health and disease. Compr Rev Food Sci Food Safety. 2019; doi: 10.1111/1541-4337.12424; Melnik BC, Kakulas F, Geddes DT, Hartmann PE, John SM, Carrera-Bastos P, Cordain L, Schmitz G. Milk miRNAs: simple nutrients or systemic functional regulators? Nutr Metab (Lond). 2016;13:42; Agrawal AK, Aqil F, Jeyabalan J, Spencer WA, Beck J, Gachuki BW, Alhakeem SS, Oben K, Munagala R, Bondada S, Gupta RC. Milk-derived exosomes for oral delivery of paclitaxel. Nanomedicine. 2017;13(5):1627-1636).

Material and Methods

Sample collection

As already outlined under general comments, early colostrum EVs and exosomes appear to represent a very unique composition to manage early stages of postnatal maturation and development of the newborn mammal. Early colostral exosomes differ substantially from mature milk exosomes which are the components of concern for human health (Melnik BC, Schmitz G. Exosomes of pasteurized milk: potential pathogens of Western diseases. J Transl Med. 2019;17(1):3).

The time points for calf plasma sampling are not based on available kinetic data of colostrum exosome uptake. It might be possible that the peak of maximal uptake lays in the interval between 6 h and 9-12 h, and may be missed in this study. The ruminant situation may change exosome uptake characteristics.

The investigators removed the fat layer, which is a major source of miRs, which thus escaped detection and follow up in this study.

The authors use the term “skim milk” throughout their manuscript, which is not correct: they analyzed “skimmed colostrum”, which should be used.

The authors state that sucrose gradient fractions of 40 % and 50 %, corresponding to a density of 1.1764 g/ml and 1.2296 g/ml, respectively, were pooled to increase yield, as preliminary results had shown these to be most enriched in colostrum EVs (data not shown).

These data are most important and should be provided.

The authors only rely on EV separation by density gradient centrifugation and ultracentrifugation. There is no control by other methods such as ExoQuick precipitation. Yamada et al. reported that the highest yield of exosomes was achieved using ultracentrifugation with ExoQuick precipitation (Yamada T, Inoshima Y, Matsuda T, Ishiguro N. Comparison of methods for isolating exosomes from bovine milk. J Vet Med Sci. 2012;74(11):1523-5). Thus, the investigators may have lost exosomes in their preparations.

Results

RNA analysis and bioinformatics

The major concern of this study is the fact that the authors do not provide absolute concentrations of their miR determinations but instead provide relative changes (delta values). Furthermore, there are no data for sensitivity and confidence intervals. Genomic mapping would be helpful to interpret the origin of the detected RNA sequences. A critical look to the data shows that regulatory miRs have been detected in EVs including tRNA, short regulatory RNAs, long non-coding RNAs and precursor-sequences of miRs.

From the point of particle energetics, it is not conceivable that exosome proteins and their miR content are disintegrated and send to different metabolic of signaling routes.

Therefore, the revised version of this manuscript should provide absolute data of miRs including a revision of corresponding figures and tables.

Discussion

The revised version of this paper should emphasize the outlined limitations of their study.

Taken together, in the present form, this paper does not allow an approximation of the proportions of colostrum EVs of exosomal versus non-exosomal origin. The miR-enriched density fraction is obviously a mixture of denser non-exosomal EVs and smaller CD63-positive colostrum exosomes. Unfortunately, the investigators used CD63 as the only exosome marker. The reviewer agrees to the conceptual idea of partly repackaging of non-exosomal EVs in intestinal mucosa cell, but not for exosomes, which are able to pass intestinal and endothelial cell boundaries.

6. PLOS authors have the option to publish the peer review history of their article (what does this mean?). If published, this will include your full peer review and any attached files.

Reviewer #1: No

---

## [Author Response · Author response to Decision Letter 0]

9 Jan 2020

A complete answer to all concerns and comments raised by the editor and reviewer can be found in our "Response to the Editor" file.

---

## [Editor Report · Decision Letter 1]

11 Feb 2020

Postprandial transfer of colostral extracellular vesicles and their protein and miRNA cargo in neonatal calves

PONE-D-19-23560R1

Dear Dr. Kirchner,

We are pleased to inform you that your manuscript has been judged scientifically suitable for publication and will be formally accepted for publication once it complies with all outstanding technical requirements.

With kind regards,

Pierre Busson, MD, PhD, Res Director

Academic Editor

PLOS ONE
---

## [Editor Report · Acceptance letter]

14 Feb 2020

PONE-D-19-23560R1 

Postprandial transfer of colostral extracellular vesicles and their protein and miRNA cargo in neonatal calves 

Dear Dr. Kirchner:

I am pleased to inform you that your manuscript has been deemed suitable for publication in PLOS ONE. Congratulations! Your manuscript is now with our production department. 

With kind regards,

on behalf of

Dr. Pierre Busson 

Academic Editor

PLOS ONE